**Subject Category:**
Biology (whole organism)

behaviour/ecology/acoustics

acoustic behaviour, passive acoustic monitoring, vocalizations, diel variation, seasonality, right whales

**Author for correspondence:**
T. A. Webster
e-mail: science@yeptrust.org.nz

# Temporal variation in the vocal behaviour of southern right whales in the Auckland Islands, New Zealand

T. A. Webster[1,2], S. M. Van Parijs[3], W. J. Rayment[1] and S. M. Dawson[1]

[1]Department of Marine Science, University of Otago, 310 Castle Street, Dunedin 9016, New Zealand
[2]Otago Museum, 419 Great King Street, Dunedin 9016, New Zealand
[3]Protected Species Branch, Northeast Fisheries Science Center, 166 Water Street, Woods Hole, MA 02543, USA

(iD) TAW, 0000-0002-3507-1162

Autonomous recorders are frequently used for examining vocal behaviour of animals, and are particularly effective in remote habitats. Southern right whales are known to have an extensive acoustic repertoire. A recorder was moored at the isolated sub-Antarctic Auckland Islands for a year to examine whether the acoustic behaviour of southern right whales differed seasonally and throughout the day at their main calving ground in New Zealand. Recordings were made in each month except June, and vocalizations were audible in all months with recordings except January. A total of 35 487 calls were detected, of which *upcalls* were the most common (11 623). Call rate peaked in August (288 ± 5.9 [s.e.] calls/hour) and July (194 ± 8.3). Vocal behaviour varied diurnally with highest call rates detected at dusk and night, consistent with the concept that *upcalls* function primarily as contact calls. Zero-inflated model results confirmed that seasonal variation was the most important factor for explaining differences in vocal behaviour. An automated detector designed to expedite the analysis process for North Atlantic right whales correctly identified 80% of *upcalls*, although false detections were frequent, particularly when call rates were low. This study is the first to attempt year-round monitoring of southern right whale presence in New Zealand.

## 1. Background

Animals vary their behaviour in response to a wide range of factors, including changes in light levels [1], availability of prey

[2], predation pressure [3,4] and weather conditions [5,6]. Similarly, vocal behaviour of many animals changes temporally. For example, terrestrial species, such as wolves (*Canis lupus* [7]) and many temperate and tropical bird species [8], exhibit diurnal variation in vocal activity with higher call rates during the late evening and dawn. The drivers of temporal variation in vocal behaviour can be difficult to determine, but may relate to optimal ambient noise and propagation conditions [9], energy reserves [10], predation risk [11] or sociality [8,12]. The calling activity of frogs [13] and some insects [14,15] is correlated with precipitation and temperature.

Marine mammals in general, and cetaceans in particular, are highly vocal, relying on sound for communication [16], navigation [17] and feeding [18]. Consequently, the occurrence of species-specific vocalizations can reveal information about a species' presence. For example, in the Gulf of Alaska, echolocation clicks made by sperm whales (*Physeter macrocephalus*) are detected more often in summer than in winter, consistent with known seasonal migration of reproductive males from productive high latitudes in summer, to the lower latitudes favoured by breeding females in winter [19]. Vocalization types and rates can change over short timescales (hours/days) or longer monthly or seasonal timescales revealing details of a species' behaviour. Blue whales (*Balaenoptera musculus*) from the eastern tropical Pacific show higher vocalization rates between February and June–thought to be associated with their foraging behaviour which is also highly seasonal [20]. Diel changes in acoustic behaviour have been observed in both mysticetes [21,22] and odontocetes [23,24]. For example, sound pressure levels of vocalizations of wintering humpback whales (*Megaptera novaeangliae*) off Maui increase at sunset, remain relatively high through the night and progressively decrease from sunrise [25]. Vocal behaviour can also be used to infer the level and type of activity that an animal is engaged in. Elevated echolocation rates and buzzes (associated with feeding) indicate that foraging at night is the norm for several odontocete species, including harbour porpoise (*Phocoena phocoena* [26]), finless porpoise (*Neophocoena asiaorientalis* [24]) and Risso's dolphin (*Grampus griseus* [23]).

Passive acoustic monitoring (PAM) has become an increasingly important research tool, particularly for documenting a species' presence [19,21] and how vocal behaviour changes with time [20]. For example, autonomous recording systems used to detect species-specific calls and monitor temporal variation in vocal behaviour have been fundamental to the establishment of protection measures for North Atlantic right whales (NARWs, *Eubalaena glacialis*) in and around shipping lanes [27]. Typically, passive acoustic detection and localization of right whales uses *upcalls* [28,29] which are the most common call type [30–32], and are produced by both males and females [33,34]. PAM is increasingly used to monitor right whales, particularly in remote regions or those with inhospitable weather conditions [35–37]. Consequently, automated detection systems [28] have been developed to facilitate processing the large quantities of data produced by PAM.

Southern right whales (SRWs, *Eubalaena australis*) have a circumpolar distribution in the Southern Hemisphere, with major breeding populations in South Africa, Argentina, Brazil, Australia and New Zealand. SRWs were hunted to the brink of extinction in New Zealand [38], but since the cessation of whaling the population has grown at an estimated 7% resulting in a recent abundance estimate of 2169 (95%: 1836–2563) [39]. For other SRW populations mainland calving grounds mean that they are easily observable. SRWs in New Zealand only calve in the remote sub-Antarctic Auckland Islands, so we know very little about distribution and behaviour through the year. Most knowledge of the behaviour of SRWs in New Zealand comes from visual surveys and observations at the sub-Antarctic Auckland Islands [40–43]. Right whales are the only cetacean species known to be regularly present in Auckland Island's waters [44] and arrive there each winter to calve from offshore feeding grounds [40]. Non-breeding whales are also present, potentially due to social factors [43]. Practical constraints restrict the duration of research expeditions to these remote islands; beyond a wintertime window of up to eight weeks [41] little is known about how or when right whales use this habitat. The vocal repertoire of SRWs at the Auckland Islands has received attention recently [45], but we do not know how long SRWs are present in this habitat or how their vocal behaviour varies over time. Given the limitations of the remote location and inclement weather conditions in the sub-Antarctic, autonomous acoustic recorders [46] present a robust tool for understanding the seasonal and diurnal calling behaviour of SRW.

This research addressed three specific questions: (1) what is the seasonal variation in the presence of whales at the Auckland Islands as revealed by autonomous acoustic recorders, (2) how do call rates and types vary temporally, both seasonally and diurnally and (3) how effectively does an automated *upcall* detector developed for North Atlantic right whales [29] perform for this population of whales?

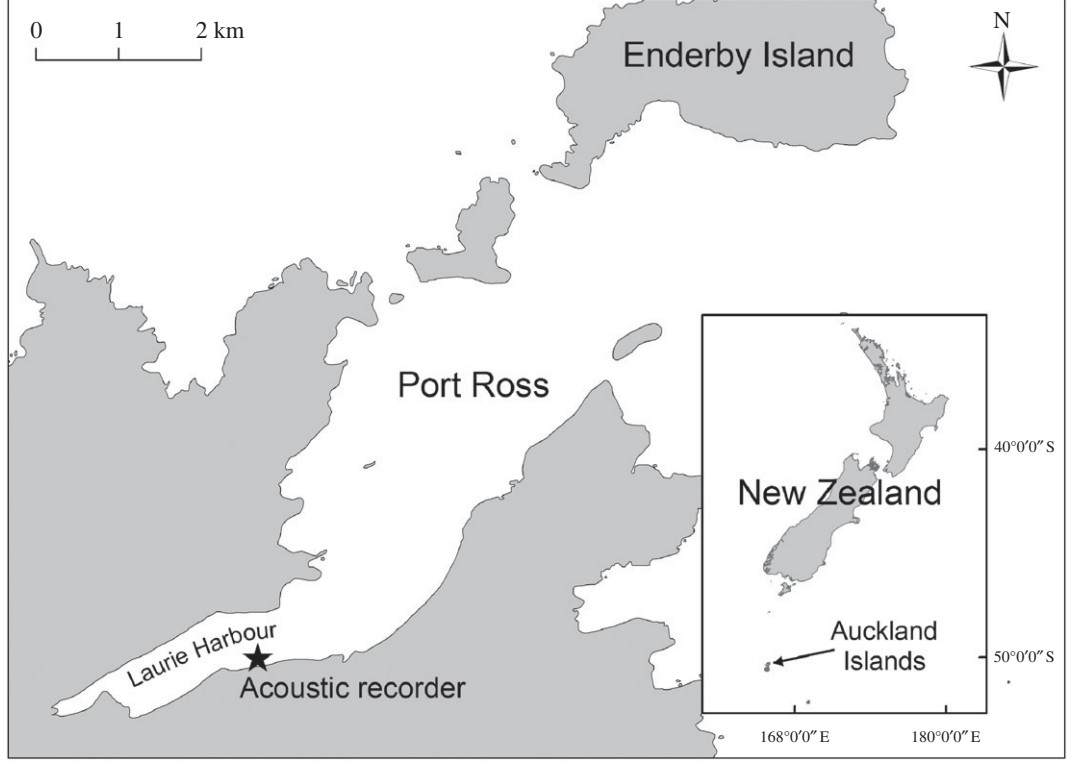

**Figure 1.** Location of the DSG-Ocean recorder for southern right whale vocalizations in Laurie Harbour, Port Ross, sub-Antarctic Auckland Islands.

# 2. Methods

## 2.1. Data collection

Recordings were made using a DSG-Ocean autonomous recorder (Loggerhead Instruments, USA, www. loggerhead.com) moored in Laurie Harbour in Port Ross, a large, sheltered harbour at the northern end of the Auckland Islands (figure 1), where SRWs occur at highest densities [42]. The deployment location, in the upper reaches of Port Ross (50°33.4′ S, 166°12.3′ E) was chosen for its physical characteristics: shallow depth (16–20 m), flat and sandy seabed, shelter from prevailing swells and minimal risk of fouling due to vessels anchoring.

The DSG-Ocean is a digital recorder with a single HTI-96-MIN hydrophone (sensitivity: −185.6 dBV/ μPa, frequency response: 20 Hz–50 kHz ± 3 dB) and programmable sampling rate and recording schedule. DSG-Ocean recorders have been used extensively for acoustic studies on fish [47–49] and are increasingly used for marine mammal research [50].

A compromise between sampling rate and recording schedule was required to maximize data quantity, given limited battery power, over an extended deployment period. A trial deployment was made between 22 July and 1 August 2011 to verify that the recorder worked and assess the practicality of the mooring system. During the test period, the DSG-Ocean recorded at a default sampling rate of 50 kHz, recording for 2.5 min every 30 min. The first long-term deployment was from 6 August 2011 to 27 July 2012. The recorder was set to a sampling rate of 4 kHz (covering the vocalization range of this species), recording for 3.75 min every 30 min (i.e. a 12.5% duty cycle).

The mooring system was designed without surface buoys or trailing lines to minimize entanglement risk. The recorder was attached via a 1 m length of 5 mm nylon line (designed to break in the event of an entanglement) to an approximate 50 kg steel mooring weight on the seabed. A second mooring weight was connected to the first via 40 m of negatively buoyant 16 mm nylon rope. The weights were deployed so the rope stretched taut along the seabed. A flotation collar attached to the top of the PVC housing of the recorder ensured that it remained upright during deployment. The recorder was retrieved by snagging the line between the two moorings with a grapnel and winching it on board.

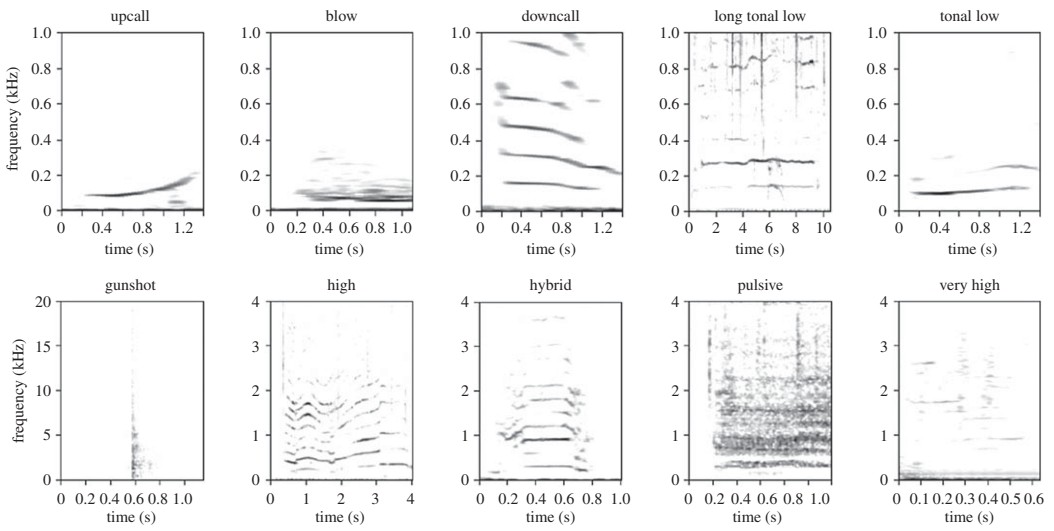

**Figure 2.** Spectrograms of 10 vocalization types assigned to southern right whales at the Auckland Islands. N.B.: time and frequency scales differ for some spectrograms. Long tonal low calls are a minor component of the vocal repertoire and have been included here for completeness, although they did not feature in the recordings analysed in this manuscript.

## 2.2. Manual vocalization detection

Acoustic data were first analysed manually using Raven Pro v. 1.5 (Bioacoustics Research Program, Cornell University, www.birds.cornell.edu). All SRW vocalizations in each of the recordings were highlighted and counted regardless of signal-to-noise ratio. Each vocalization was assigned to one of nine types (*upcall*, *downcall*, *gunshot*, *high*, *pulsive*, *tonal low*, *very high*, *hybrid* or *blow*; figure 2) based on a previously determined quantitative classification system described in a concurrent study of this population [45].

## 2.3. Automated vocalization detection

All acoustic recordings were also examined using an automated detector [29] in the eXtensible BioAcoustic Tool (XBAT [51]) written in MATLAB (The Mathworks Inc., MA, USA). The automated call detection algorithm is specifically designed to detect *upcall* vocalizations produced by NARWs [31]. The XBAT detector was set at a correlation threshold of 0.4, a value typically used for detections of NARW calls. Acoustic similarity between a call template and possible *upcall* events was assessed using a Generalized Likelihood Ratio Test based scheme, and all events exceeding this threshold were logged.

## 2.4. Verification of detector results

Manual and automated detections were compared in order to test the effectiveness and sensitivity of the algorithm for detecting SRW *upcalls*. We randomly selected 10% ($n = 1443$) of the total number of recordings for verification analysis which included representation of each month. Each recording was reviewed in XBAT using the *event palette* to display all detection events logged by the detector. The detection events were then annotated as either true or false *upcall* detections based on the manual classification.

Two measures of detector performance were applied [52]. The first (Positive Predictive Value; PPV) represents the percentage of the *upcalls* logged by the detector that were judged by the human analyst to have been correctly assigned, and the converse accounted for false positives. The second measure (True Positive Rate, TPR) is the percentage of *upcalls* detected by the analyst also logged by the detector, and the converse enabled estimation of false negatives.

Deviation of the automated detections from the number of manually detected *upcalls* was examined using regression. In this context, the slope of the fitted line indicated bias (1 : 1 = zero bias), while the fit of the line indicated the consistency of assignment.

## 2.5. Statistical analyses

Acoustic recordings were examined for any seasonal or diel patterns in vocalizations. Data on the timing of sunrise, sunset, transit time and nautical dusk/dawn at Port Ross were obtained from Land Information New Zealand (LINZ, www.linz.govt.nz). Transit was defined as the maximum altitude of the sun relative to the horizon. Nautical twilight was defined as the period between nautical dawn (or dusk), when the sun is 12° below the horizon, and sunrise (or sunset). Each acoustic recording was assigned to one of five diel periods: dawn (nautical dawn to sunrise), morning (sunrise to transit), afternoon (transit to nautical dusk), dusk (sunset to nautical dusk) and night (nautical dusk to nautical dawn). The duration of each diel period varied between season, with longer daylight hours in summer (mid-summer: dawn ≈2 h, day ≈14 h, dusk ≈2 h, night ≈6 h) than winter (mid-winter: dawn ≈1.5 h, day ≈8 h, dusk ≈1.5 h, night ≈13 h). Recordings were also classified by season; December to February was defined as Summer, March to May as Spring, June to August as Winter and September to November as Autumn.

All statistical analyses were carried out in R v. 3.0.2 (R Development Core Team 2004, www.R-project. org). To assess seasonal and diel variation in vocal behaviour, hourly calling rates (all calls) and hourly *upcall* rates were calculated from the manual call appraisal. Diel patterns were examined by comparing rates during hour-long time bins based on local time (e.g. 00.00–01.00, 01.00–02.00) and diel periods (dawn, morning, afternoon, dusk, night). Call rates were adjusted to correct for the variation observed during the year in terms of total number of calls detected each day, i.e. to remove any influence of times with very high call rates. The corrections were calculated by subtracting the overall calling rate (calls/$h^{-1}$) for each day from the hourly calling rate within a particular period (e.g. hour or diel period) [53–55]. This correction resulted in a mean adjusted call rate, where positive values indicate an above average (and negative a below average) rate during a particular time period allowing for seasonal variation in the total number of calls detected. A Zero Inflated Model (ZIM) with a negative binomial distribution was used to evaluate whether particular seasons and/or diel periods were important for explaining differences in right whale call presence and call rate. A ZIM is a two component mixture model [56] and was used in preference to a Generalized Linear Model (GLM) because it accounted for the false zeros which may have been present due to whales being absent from the study area [57]. The two components are a binomial GLM which models the probability of obtaining a false zero, and a negative binomial GLM which models the count data [57]. A log-likelihood ratio test [58] was used to determine whether a poisson or negative binomial distribution best fitted the count data.

# 3. Results

The acoustic recorder was deployed at the Auckland Islands for 347 days from 22 July 2011 during which there were a total of 302 recording days (80.5% of deployment days), resulting in 892 h of acoustic data. No recordings were made after 24 May 2012 when the batteries ran out.[1] The recorder was out of the water from 2 to 5 August 2011 inclusive. A total of 35 487 right whale calls were manually detected during the deployment period.

## 3.1. Seasonal variation

Right whale vocalizations were detected in all months with recordings except for January. Vocalization rates varied dramatically during the deployment period (figure 3) and showed a strong seasonal trend. Vocalization rates were highest in the austral winter (219 ± 4.9 calls $h^{-1}$ ± s.e.) and lowest during the summer (0.2 ± 0.06). A similar seasonal pattern was evident when *upcalls* were examined separately (figure 3).

The months with the highest call rates were August (288 ± 5.9 calls $h^{-1}$) and July (194 ± 8.3). Mean vocalization rates remained relatively high in September (91 ± 3.6) before declining through October (27 ± 7.2) and November (6 ± 0.7). No calls were observed in January and although calls were detected in December, February, March and April, the mean rate was very low (less than 1 call $h^{-1}$). The vocalization rate increased sharply in May (41 ± 2.9).

---

[1]There were two minor gaps in data due to software bugs. Recordings were skipped between 01.30 and 03.00 on 25 September 2011 due to daylight saving, and the entire day on 1 March 2012 due to a leap year.

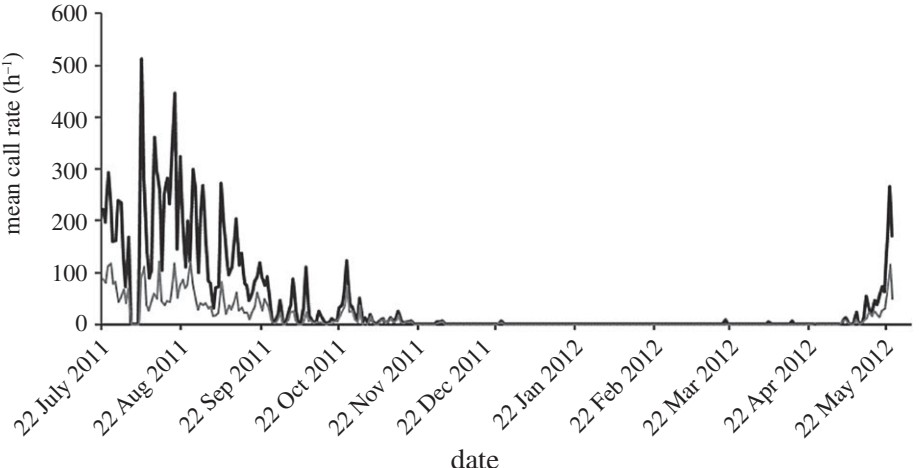

**Figure 3.** Mean number of all calls (black) and *upcalls* (grey) per day for southern right whales at the Auckland Islands. N.B.: no data were collected from 2 to 5 August 2011 (inclusive).

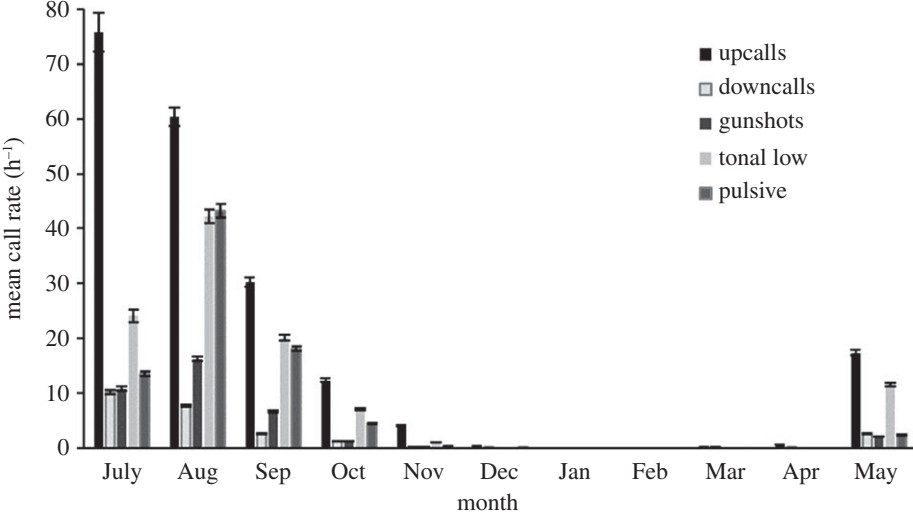

**Figure 4.** Mean call rate (calls h$^{-1}$ $\pm$ s.e.) for the five most common call types (*upcall*, *downcall*, *gunshot*, *tonal low* and *pulsive*) in each month for southern right whales recorded at the Auckland Islands between July 2011 and May 2012.

*Upcalls* were evident in all months that vocalizations were detected and *upcall* rate followed a broadly similar trend to all vocalizations (figure 3). *Upcalls* were the most prevalent of the vocalization types, 11 623 (33%) of all of the calls detected. These vocalizations typically accounted for a smaller proportion of calls, 30–40%, between July and October when the overall call rate was high, and rose to 93% in April and 100% in February when vocalization rates were low. Mean *upcall* vocalization rates were highest in July (76 $\pm$ 4.1 calls h$^{-1}$) and August (60 $\pm$ 2.2).

Other vocalization types (*downcall*, *gunshot*, *high*, *hybrid*, *very high* and *blow*; [45]) were recorded much less frequently than *upcalls*, and in some months not at all (particularly during summer). Collectively, however, these vocalization types accounted for 67% of the total calls observed. *Hybrid*, *high* and *gunshot* vocalizations exhibited similar patterns of seasonal presence and were completely absent between December and April. *Very high* calls were only detected during a four-month period between July and October and at very low levels (less than 1 call h$^{-1}$). Seasonal patterns in vocal behaviour of SRWs for the five most abundant call types (*upcall*, *tonal low*, *pulsive*, *downcall* and *gunshot*) showed that call rate for each call type was highest during winter months and lowest during summer months (figure 4). The proportion of the five most abundant call types, however, varied little over the year; *upcall* rates were highest, then *tonal* and *pulsive* calls, and *gunshots* and *downcalls* were always the lowest (figure 4).

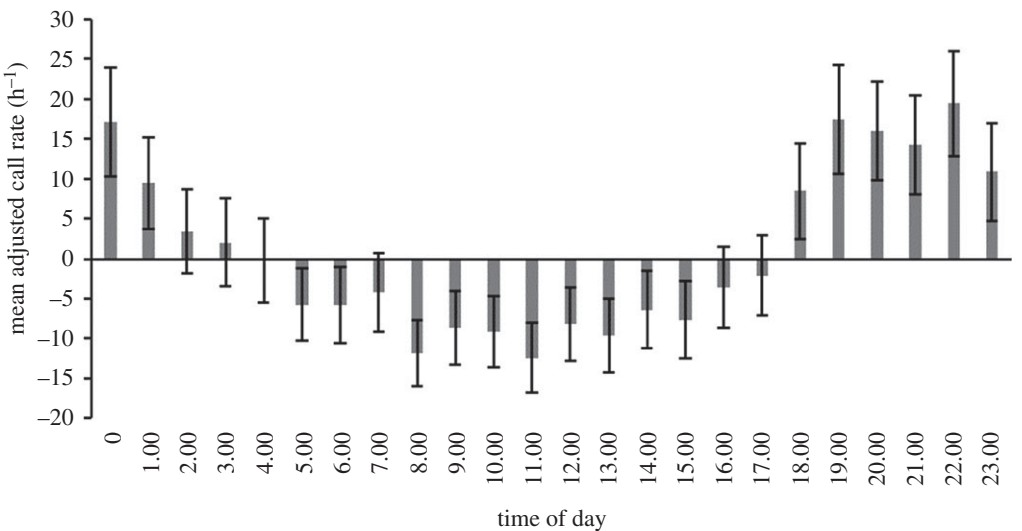

**Figure 5.** Mean adjusted hourly vocalization rate (calls h$^{-1}$ ± s.e.) for southern right whales recorded at the Auckland Islands between July 2011 and May 2012. Rates are adjusted to compensate for variation in call rate throughout the year. Positive values indicate an above average rate and negative values indicate a below average rate.

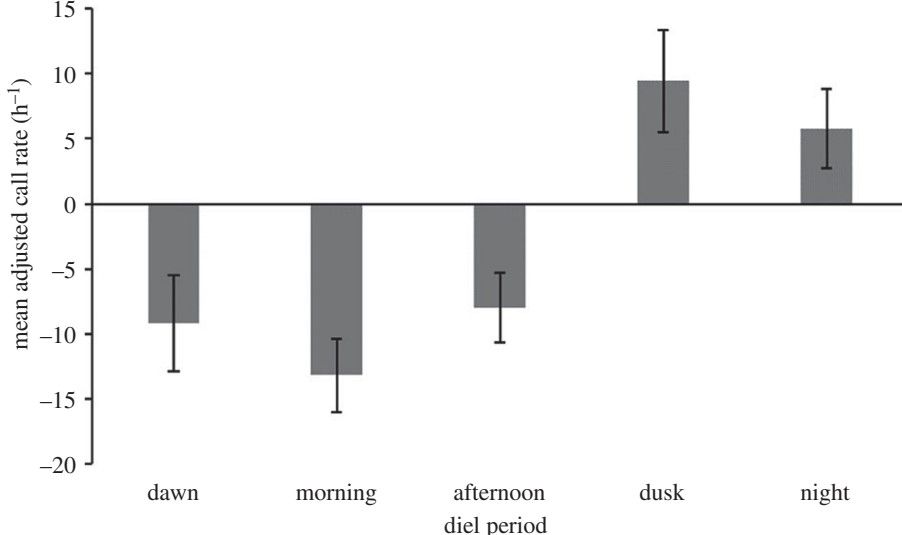

**Figure 6.** Mean adjusted call rate (calls h$^{-1}$ ± s.e.) per diel period for southern right whales recorded at the Auckland Islands between July 2011 and May 2012. Rates are adjusted to compensate for variation in call rate throughout the year. Positive values indicate an above average rate and negative values indicate a below average rate.

## 3.2. Diel variation

While vocalizations were recorded during all hours of the day and night a strong diurnal pattern was evident (figures 5 and 6). Mean vocalization rate increased during the evening and remained high throughout the night. Mean adjusted call rates exhibited a similar diel pattern to mean call rates and accounted for the fact that call rates were higher during winter months (figure 6). The same diel pattern was evident with higher call rates at dusk and night when *upcall* rate was examined separately.

Diel patterns in call rate for the five most abundant call types (*upcall, tonal low, pulsive downcall* and *gunshot*) varied throughout the day with highest rates at dusk and during the night (figure 7). There was no time of day when particular call types predominated; calls were used in broadly similar proportions within each diel period (figure 7).

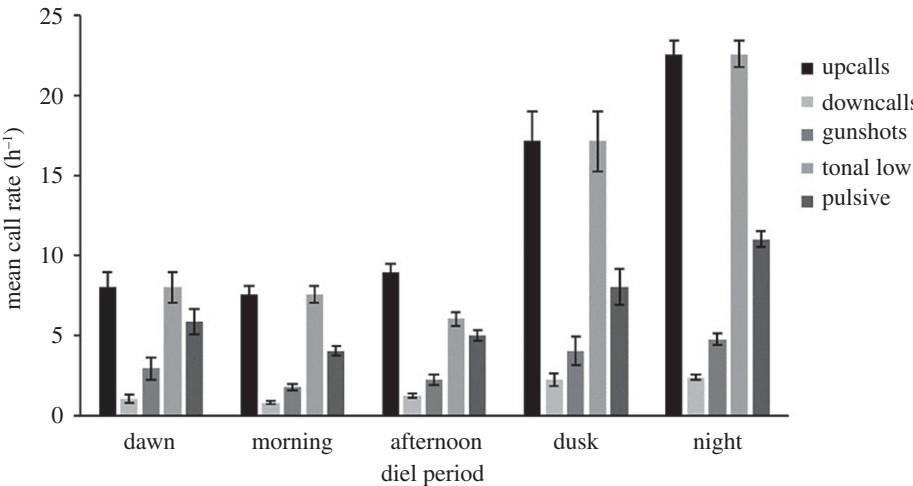

**Figure 7.** Mean call rate (calls h$^{-1}$ ± s.e.) for the five most common call types (*upcall*, *tonal low*, *pulsive downcall* and *gunshot*) in each diel period for southern right whales recorded at the Auckland Islands between July 2011 and May 2012.

## 3.3. Modelling of effects on call presence and rate

A log-likelihood ratio test revealed that a negative binomial model best fitted the count data. The ZIM results revealed a strong effect of season on the presence and acoustic behaviour of whales, and a lesser effect of diel period (table 1). The ZIM coefficients showed that false zeros were most likely to occur in summer, suggesting whales were more likely to be absent from the study area. The count model coefficients suggest that call rate was significantly higher in winter, when whales were more likely to be present. The count data also show that call rates were higher during dusk and night-time periods and lower in the morning.

## 3.4. Effectiveness of automated detector

The automated detector designed to identify NARW *upcalls* [29] worked well for detecting SRW *upcalls* at the Auckland Islands. There was a positive relationship between the manually observed and automatically detected *upcalls*, although the automated detector picked out slightly more calls than the human analyst (figure 8 and table 2). The TPR (sensitivity) of the detector was high; 80.4% (s.e. ± 1.7) of manually categorized *upcalls* were identified by the detector. Thus, the rate of missed calls was around 20%. The mean PPV of the detector was, however, relatively low with only 40.4% (s.e. ± 1.9) of detected *upcalls* correctly identified. Mean PPV was higher (47–67%) when monthly vocalization rates were high, but dropped between October and April to a mean of 16% when vocal activity was minimal. Therefore, many false detections were made by the automated system, particularly when call rate was low. A large proportion of the false detections, however, were other right whale call types (e.g. part of a *hybrid* call). These were classified as false detections here because they were not strictly *upcalls*.

# 4. Discussion

## 4.1. Seasonal variation

This study found that SRWs are present more often and for far longer periods than previously observed at the Auckland Islands [41]. Call rates indicate that right whales are vocally active (and therefore routinely present) between May and November. Except for January, vocalizations were detected in every month of the year, indicating at least occasional presence in these months. Previously, SRWs were thought to be present at the Auckland Islands from early May to the end of September [41]. Long-term acoustic studies regularly find that marine mammals are present far more frequently than observed visually [37,59]. Peak vocalization rates (July–August) occurred in the same months as previous visual observations of peak abundance [41]. If SRWs are only passing through or using the outer part of the harbour it is possible that they would not be detected due to the location of the

**Table 1.** Results of the zero-inflated model examining seasonal and diel differences in *all call* and *upcall* presence and rate for right whales recorded at the Auckland Islands. The reference level for diel period is afternoon and the reference level for season is autumn. Significant results are in italics.

| variable | all calls | | | | upcalls only | | | |
|---|---|---|---|---|---|---|---|---|
| | estimate | s.e. | z-value | Pr(>\|z\|) | estimate | s.e. | z-value | Pr(>\|z\|) |
| *count model coefficients* | | | | | | | | |
| season spring | 0.05 | 0.05 | 0.92 | 0.36 | 0.08 | 0.06 | 1.43 | 0.15 |
| season summer | −0.54 | 0.28 | −1.91 | 0.06 | −0.19 | 0.27 | −0.69 | 0.49 |
| season winter | 0.64 | 0.05 | 12.69 | $<2.00 \times 10^{-16}$ | 0.26 | 0.06 | 4.76 | $1.92 \times 10^{-6}$ |
| diel period dawn | 0.02 | 0.08 | 0.32 | 0.75 | −0.14 | 0.09 | −1.61 | 0.11 |
| diel period dusk | 0.28 | 0.07 | 3.99 | $6.52 \times 10^{-5}$ | 0.30 | 0.08 | 3.79 | $1.48 \times 10^{-4}$ |
| diel period morning | −0.13 | 0.05 | −2.62 | *0.01* | −0.01 | 0.06 | −0.26 | 0.79 |
| diel period night | 0.25 | 0.04 | 6.27 | $3.66 \times 10^{-10}$ | 0.24 | 0.04 | 5.34 | $9.33 \times 10^{-8}$ |
| *zero-inflation model coefficients* | | | | | | | | |
| season spring | −1.61 | 0.06 | −24.85 | $<2.00 \times 10^{-16}$ | −1.32 | 0.07 | −18.19 | $<2.00 \times 10^{-16}$ |
| season summer | 3.65 | 0.32 | 11.35 | $<2.00 \times 10^{-16}$ | 3.35 | 0.32 | 10.39 | $<2.00 \times 10^{-16}$ |
| season winter | −4.80 | 0.11 | −44.68 | $<2.00 \times 10^{-16}$ | −3.49 | 0.08 | −42.20 | $<2.00 \times 10^{-16}$ |
| diel period dawn | 0.00 | 0.13 | −0.02 | 0.99 | −0.01 | 0.14 | −0.06 | 0.95 |
| diel period dusk | −0.47 | 0.12 | −3.79 | $1.49 \times 10^{-4}$ | −0.49 | 0.13 | −3.93 | $8.62 \times 10^{-5}$ |
| diel period morning | 0.13 | 0.08 | 1.63 | 0.10 | 0.15 | 0.08 | 1.83 | 0.07 |
| diel period night | −0.66 | 0.07 | −9.50 | $<2.00 \times 10^{-16}$ | −0.68 | 0.07 | −9.56 | $<2.00 \times 10^{-16}$ |

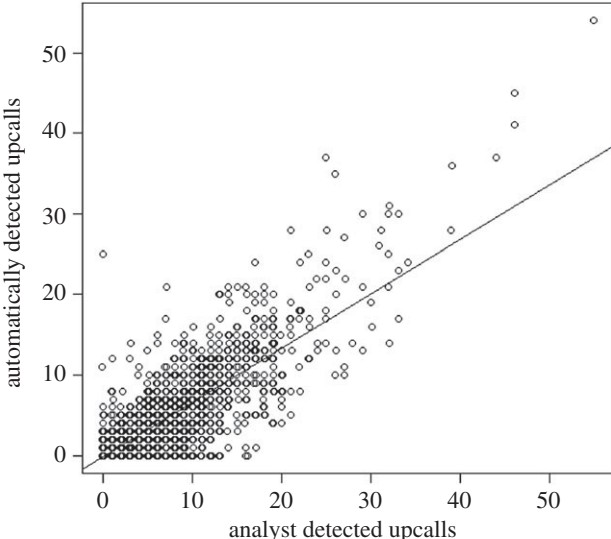

**Figure 8.** The number of automatically detected *upcalls* per recording versus the number of *upcalls* detected by the analyst. Right whale vocalizations were recorded at the Auckland Islands between July 2011 and May 2012. Best-fit linear regression is shown ($y = 0.6743x - 0.1261$, $r^2 = 0.76$). Points have been jittered by 0.6 to offset and reveal overlapping points.

recorder in the inner part of Port Ross. Also, to be recorded, the whales have to be calling. For these reasons, vocal measures of presence should be considered conservative indicators of SRW presence at the Auckland Islands. It should also be noted that the long periods of silence, in January for example, are presumed to indicate that whales are not present at the Auckland Islands, rather than a change of behaviour that has resulted in the cessation of vocalizations. This is consistent with what is known of seasonal migration behaviour of southern right whales, i.e. offshore foraging in summer and inshore calving in winter [60]. Comparative analyses between autonomous passive acoustic and visual techniques using a solar-powered camera in the Auckland Islands showed that both techniques were comparable for monitoring southern right whale presence [61]. However, while the solar-powered camera was able to record information over a longer period than the battery-limited acoustic recorder, the acoustic recorder was less limited by weather conditions, sea state and light availability.

The vocalization rate in winter was four times that of any other season. Consequently, the ZIM results showed that seasonal variation was the biggest contributing factor to call presence. Strong seasonal trends in call occurrence have been observed in other populations of right whales [37,54]. Seasonal variability in call detection is unsurprising given that right whales migrate between feeding and calving habitats [60]. Taken together, however, the acoustic studies show that right whales are present in particular habitats more often than expected given what was previously understood of their movements.

*Upcalls* were the most abundant call type detected (33%) and remained so throughout the year. Recent research [58] on NARW vocalizations also found *upcalls* to be the most frequently used call type, although they constituted a higher proportion (49%) of all calls than in this study. This variation in proportion could be the result of comparing two different (albeit closely related) species or because whales were sampled in different habitats. Given that *upcalls* appear to function as contact calls [30,62] it makes sense that they are the most commonly produced call type. *In situ* recordings made in July and August indicate that *upcalls* are made similarly often to *tonal low* and *pulsive* calls [45]. This underscores the value of the much larger dataset provided via autonomous recording.

Previous studies have shown that all right whale species use specific calls in different behavioural contexts [33,34,63] and that the relative proportion of call types varies both seasonally and geographically [46]. In this study, *very high*, *hybrid*, *high* and *gunshot* calls were observed only during a period of between four to six months of the year. This potentially indicates a seasonal change in behaviour, or may simply reflect the low call rates detected outside the main winter period. *Gunshots*, *hybrid* and *high* calls are understood to be related to mating activities [34,63] and the relative proportions of these calls increase during the breeding season [46,64]. The Auckland Islands are the major calving grounds for SRWs in New Zealand waters [41,42], but it is currently unclear where most mating occurs. Despite extensive photo-identification work (Rayment WJ, 2018, unpublished

**Table 2.** Results of the manual detections (made by an analyst) compared to the automated *upcall* detections for 10% of the total number of acoustic recordings. Data are given for each month separately, except for October to April, where data were combined due to low call rates.

| month | number of upcalls manually detected | number of upcalls autodetected | number of true positive (TP) upcalls | number of false positive (FP) upcalls | number of false negative (FN) upcalls | positive predictive value (TP/(TP + FP)) | True Positive Rate (TPR) (TP/(TP + FN)) |
|---|---|---|---|---|---|---|---|
| July | 183 | 266 | 146 | 113 | 37 | 56.1 | 80.0 |
| August | 551 | 693 | 455 | 244 | 93 | 58.2 | 82.0 |
| September | 376 | 488 | 302 | 183 | 74 | 47.5 | 81.8 |
| October–April | 156 | 410 | 135 | 282 | 19 | 16.1 | 81.8 |
| May | 169 | 173 | 131 | 40 | 38 | 67.3 | 72.9 |
| overall detector effectiveness | | | | | | 40.4% | 80.4% |
| | | | | | | (95% CI: 36.6 – 44.2) | (95% CI: 76.9 – 83.8) |

data), no female right whale has been seen at the Auckland Islands in the year before giving birth. If the gestation period is about a year, as in SRWs in South Africa [65], the Auckland Islands are not likely to be the principal mating site for SRWs in New Zealand. Further acoustic monitoring and exploration of other areas such as Campbell Island [66] may help locate the main mating habitat for this population.

## 4.2. Diel variation

Call rates showed a clear diel pattern, with more vocal activity at dusk and during the night than during the day (dawn, morning and afternoon). *Upcalls* showed a similar pattern when examined separately. Mean adjusted vocalization rates, which compensated for the high winter call rates, also exhibited the same diel pattern.

The rate at which right whales call varies with call type, activity of the whales, group composition and group size [33,46,64]. Right whales call more often during twilight and night in winter calving areas [67] and on feeding grounds [54,55,64]. On feeding grounds, the behaviour of NARWs is strongly influenced by the diel migration of copepods, their primary prey [68]. This extends to their vocal behaviour, where an inverse relationship between foraging and call production has been suggested [64]. Daily activity cycles dictated by prey availability have also been observed for other mysticetes and are often linked to the vertical migration of prey [20,53]. Sei whales (*Balaenoptera borealis*) called more frequently during the day when prey was less available and hence the whales were not feeding [22]. The high rate of NARW *upcalls* during twilight periods is considered to be a reflection of prey distribution, i.e. when prey are more dispersed and foraging intensity is lower, call rate increases [55]. There is little evidence of right whales feeding at the Auckland Islands [40] and baleen whales seldom feed while on calving grounds [69]. It is therefore unlikely that vocalization rates at the Auckland Islands are much affected by prey availability. Diel patterns in vocalization rate could also indicate a change in social behaviour or a general increase in activity or communication [33]. In NARWs *gunshot* calls, potentially indicative of mating activity, are most frequent in the late afternoon and evening [70]. Chorusing by humpback whales on wintering grounds in Hawaii increases at night, both in terms of amplitude and the number of whales involved [25]. While there was little change in the relative proportions of different call types throughout the day, SRWs at the Auckland Islands vocalize more at dusk and night. This may be because reduced visual contact in the dark increases the need for acoustic communication, particularly between mother–calf pairs. Risk of predation is likely to be low for SRWs at the Auckland Islands and hence communication at night may not incur the cost it would for other species [71].

## 4.3. Detection of vocalizations

Throughout the year, *upcalls* were more prevalent than any other vocalization type and were made at all times of the day. Overall they constituted a third of the calls recorded. Particularly given that *upcalls* are produced by both males and females [33,34], they are especially appropriate for automated detection. While there is ongoing development of algorithms to detect other right whale call types [72,73], to date the focus remains on *upcalls* [28,29]. Although no call-positive days were missed by using only an *upcall* detector, the addition of other call types would increase detection probability, particularly when call rates are low.

The strong positive relationship between the manually observed and automatically detected *upcalls*, consistent throughout the year, confirms the utility of the detector. Importantly, however, the false positive detection rate of the automated detector was high, particularly during periods of low vocal activity. The detection threshold of the algorithm could be increased to reduce the number of false positives, but this would also increase the rate of false negatives (missed calls). In our study, false detections were often right whale vocalizations other than *upcalls*. In areas with other mysticetes, false detections will sometimes be species other than right whales. An additional problem is that the true number of calls is never known precisely—detector efficiency can only be compared against a human analyst, an imperfect standard at best [28].

This study has proven the effectiveness of using an autonomous passive acoustic recorder to provide detailed data on vocal behaviour of southern right whales in a remote area seldom visited by vessels. Previously, information on habitat use in this area was limited to visual observations during relatively short visits in the austral winter [40,42,43]. Autonomous recorders have compelling additional benefits: they are viable 24 h a day, are less restricted by bad weather and are much more affordable

than deploying a visual survey team in a remote location for long time periods. Furthermore, different call types might be indicative of different behaviours which may not be apparent from visual observation alone. Additionally, as demonstrated in this study, acoustic data are much more amenable to automated analysis.

In this study, we used PAM to investigate the acoustic ecology and behaviour of SRW at their only known calving ground in New Zealand. SRWs in particular lend themselves to PAM due to their highly vocal nature and extensive vocal repertoire. The year-round monitoring of SRWs at the Auckland Islands expanded upon our current knowledge, revealing that they are at least occasionally present in every month of the year. Call rates in winter were, however, four times higher than any other season, suggesting seasonal variations in density as a result of winter migration inshore to calve. *Upcalls* were the most commonly produced call type proving ideal for automated detection throughout the year. Seasonality was apparent in many other call types, likely indicating a change of behaviour during winter. Call rates exhibited a clear diurnal pattern with higher vocal activity during dusk and night-time, potentially because reduced visual contact increases the need for acoustic communication. This study begins to scratch the surface of temporal variation in the acoustic behaviour of SRWs, but there is much more to be revealed.

Ethics. Research was conducted under Marine Mammal Permit Per/NO/2010/05 issued by the Department of Conservation to WR and SD. The acoustic recorder was deployed in Port Ross in compliance with a Permit SO-31108-MAR issued by the Department of Conservation to TW.

Data accessibility. Raw data and R scripts can be found in the electronic supplementary material which has been submitted to Dryad Digital Repository: http://dx.doi.org/10.5061/dryad.ps58g58 [74].

Authors' contributions. T.W., S.D. and S.V.P. conceived and designed the study. T.W. collected the field data and conducted the data analyses. S.V.P. provided the automated detector and assisted with its set-up. T.W. conducted statistical analyses, with advice from W.R. T.W. prepared the draft manuscript and review comments were provided by S.D., W.R. and S.V.P. All authors gave final approval for publication.

Competing interests. The authors declare that they have no competing interests.

Funding. This project was funded by the Foundation for Research Science and Technology, Otago Museum, University of Otago and the New Zealand Whale and Dolphin Trust. T.W. was supported by a PhD scholarship and publishing bursary from the University of Otago.

Acknowledgements. Bill Dickson, Phil Heseltine, Steve Little, Evan Kenton and Steve King expertly crewed *Polaris II* and supported research operations in the field. David Mann provided advice and assistance with acoustic recorders at all stages and Glen Rowe from Land Information New Zealand provided essential data on nautical twilight, sunrise and sunset times. Thanks to the staff at the Marine Science Department at University of Otago for logistical support, and Department of Conservation staff, including Pete McLelland, Doug Veint, Jo Hiscock and Gilly Adam for assistance with permitting and quarantine procedures.

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
