## [Reviewer comments · Royal Society Open Science]

Review History

RSOS-181487.R0 (Original submission)

Review form: Reviewer 1 (Amy Knowlton)

Is the manuscript scientifically sound in its present form?

Yes

Are the interpretations and conclusions justified by the results?

Yes

Is the language acceptable?

Yes

Is it clear how to access all supporting data?

Yes

Do you have any ethical concerns with this paper?

No

Have you any concerns about statistical analyses in this paper?

I do not feel qualified to assess the statistics

Recommendation?

Accept with minor revision (please list in comments)

Comments to the Author(s)

This is a really nice study. My only main comment is that it would be helpful for the reader to understand how this habitat relates to the bigger picture of SRW distribution and knowledge. I've included some minor comments within the attached document (Appendix A).

Review form: Reviewer 2

Is the manuscript scientifically sound in its present form?

Yes

Are the interpretations and conclusions justified by the results?

Yes

Is the language acceptable?

No

Is it clear how to access all supporting data?

Yes

Do you have any ethical concerns with this paper?

No

Have you any concerns about statistical analyses in this paper?

No

Recommendation?

Major revision is needed (please make suggestions in comments)

Comments to the Author(s)

This manuscript examines the seasonal and diel acoustic behaviour of southern right whales at a remote calving ground using PAM. The paper jumps between focusing on PAM and SRW acoustic ecology, which leads to a lack of conceptual flow. A little more cohesion between the concepts will help the manuscript flow better (e.g., acoustic ecology of a difficult to study species is revealed by PAM). A thorough read of the manuscript is needed to ensure the grammatical errors throughout are remedied. Some (but by no means all) are noted in the specific comments below. There are too many Figures presented with the majority replicating information and can be removed from the manuscript. What is missing, however, is a clear Table detailing the auto-detector success/failure rates. After major revision the manuscript will be an important contribution to the literature.

Specific comments are provided below:

Abstract

L3-6: I suggest combining these two sentences, and adding in a sentence about southern right whales (e.g., distribution, ecology, etc.). Your opening paragraph in the Introduction focuses on behaviour and yet your abstract focuses on PAM.

L10: add “detected” after call rates.

Background

L22-30: I suggest including the word “ | diurnal” in this paragraph – as that is what you are describing.

L54: Add “For example,” autonomous recordings....

This will allow your sentences to link better.

L64: You need a link to the previous paragraph, for example: “In the Southern Hemisphere, most knowledge....”

L64-6: What about genetic studies? These have revealed information about migratory behavior, etc. Please include.

L74: “an excellent tool to answer these questions”. Reword to something a little more focused on your coming questions e.g., “provide/present a viable/robust tool for understanding the seasonal and diurnal calling behavior of SRW”. This then links into your next paragraph.

L76-80: Q1 is basically revealed by Q2 so I don’t see why this is a separate focus. PAM is important and will reveal presence. Again, is this paper focused on PAM or on the acoustic ecology of the species? My feeling is the latter and PAM is the tool to get you to the important questions about your species.

Methods

L86: delete “of”.

L103: i.e., a 12.5% duty cycle. Please add. Please also mention that a 4 kHz sampling rate covers the vocalisation range of this species.

L105-12: Please include the specific depth at the deployment location and the depth of the recorder in the water column.

L115-9: Your classification methods need to be expanded. So, you first manually scanned all recordings for call presence. Does this include all calls no matter the SNR? If so, please state this or the SNR you used as a cut off. Once a call was identified, you classified the call type based on a previous quantitative classification of a subset of your data. Please include this information. From here you then counted the number of calls and call types per recording/hour/day/month? More information is required.

L122: all acoustic recordings or a subset? Please clarify.

L136: what about missed detections? How many calls were missed by the detector? There is no Table in Results detailing number of calls manually classified, classified by detector, missed by

detector, missed by human, etc. Please add such a Table. [I see information is provided in the SI but this should be summarised for the reader in a Table].

L163: delete "Program".

L170: "calls" should be "call".

L172: More information is required about the difference between positive and negative values. "This correction resulted in a mean adjusted call rate where positive values indicate....".

Results

Footnote 1 pg 9: remove "a" from the first sentence.

L200: This is half of a sentence. I suggest including a statement after this along the lines of "The vocalisation rate increased sharply in May (41, 2.9), coinciding with[winter migration into the area]."

I would like to see a Figure with spectrograms of the five major call types for the generalist reader. Corresponding audio files would also be nice. Especially for the upcall, which you run and auto detector on. Ideally, at least 5 examples of different upcalls would be good to see what variability in call is allowable to be classified by the detector (and human as an upcall).

L245-57: This information should be presented in a clear Table outlining the detector's success (or failure). Number of calls manually detected, Number of calls autodetected, number of missed calls, number of false detections - all with actual counts and percentages. Given this is then discussed by month I suggest then breaking it down by month too. This was run on 10% of the data - was this spread across all months? Please include more information in the methods at L133 to confirm this.

Discussion

After the focus of the paper is clarified (PAM or acoustic ecology), the Discussion can be tightened to reflect the focus. I suggest adding a concluding paragraph to leave the reader with a clear take home message.

Figures and Tables

There are too many figures.

Figure 2: delete figure as the information is explained in the text and covered in Figures 3 & 4.

Delete Figure 4 as Figure 5 provides more information. If the authors feel information from Fig. 4 needs to be included, this should be presented in a Table of call numbers.

Figure 6 & 7: more information about what positive and negative numbers mean is required. Each Figure should be stand-alone and not require consultation with the main text.

Decision letter (RSOS-181487.R0)

04-Jan-2019

Dear Dr Webster,

The editors assigned to your paper ("Temporal variation in the vocal behaviour of southern right whales in the Auckland Islands, New Zealand") have now received comments from reviewers. We would like you to revise your paper in accordance with the referee and Associate Editor suggestions which can be found below (not including confidential reports to the Editor). Please note this decision does not guarantee eventual acceptance.

Please submit a copy of your revised paper before 27-Jan-2019. Please note that the revision deadline will expire at 00.00am on this date. If we do not hear from you within this time then it will be assumed that the paper has been withdrawn. In exceptional circumstances, extensions may be possible if agreed with the Editorial Office in advance. We do not allow multiple rounds of revision so we urge you to make every effort to fully address all of the comments at this stage. If deemed necessary by the Editors, your manuscript will be sent back to one or more of the original reviewers for assessment. If the original reviewers are not available, we may invite new reviewers.

- Data accessibility

If you wish to submit your supporting data or code to Dryad (<http://datadryad.org/>), or modify your current submission to dryad, please use the following link:
<http://datadryad.org/submit?journalID=RSOS&manu=RSOS-181487>

- **Competing interests**

- **Authors' contributions**

- **Acknowledgements**

- **Funding statement**

Kind regards,

Andrew Dunn

on behalf of Dr Asha de Vos (Associate Editor) and Kevin Padian (Subject Editor)

Associate Editor's comments (Dr Asha de Vos):

Associate Editor: 1

Comments to the Author:

This manuscript will be a valuable contribution to the literature after the suggested revisions.

Thank you for your submission and we hope you will be able to address the reviewers concerns in a timely manner.

Comments to Author:

Reviewers' Comments to Author:

Reviewer: 1

Comments to the Author(s)

This is a really nice study. My only main comment is that it would be helpful for the reader to understand how this habitat relates to the bigger picture of SRW distribution and knowledge. I've included some minor comments within the attached document.

Reviewer: 2

Comments to the Author(s)

This manuscript examines the seasonal and diel acoustic behaviour of southern right whales at a remote calving ground using PAM. The paper jumps between focusing on PAM and SRW acoustic ecology, which leads to a lack of conceptual flow. A little more cohesion between the concepts will help the manuscript flow better (e.g., acoustic ecology of a difficult to study species is revealed by PAM). A thorough read of the manuscript is needed to ensure the grammatical errors throughout are remedied. Some (but by no means all) are noted in the specific comments below. There are too many Figures presented with the majority replicating information and can be removed from the manuscript. What is missing, however, is a clear Table detailing the auto-detector success/failure rates. After major revision the manuscript will be an important contribution to the literature.

Specific comments are provided below:

Abstract

L3-6: I suggest combining these two sentences, and adding in a sentence about southern right whales (e.g., distribution, ecology, etc.). Your opening paragraph in the Introduction focuses on behaviour and yet your abstract focuses on PAM.

L10: add "detected" after call rates.

Background

L22-30: I suggest including the word " | diurnal" in this paragraph - as that is what you are describing.

L54: Add "For example," autonomous recordings....
This will allow your sentences to link better.

L64: You need a link to the previous paragraph, for example: "In the Southern Hemisphere, most knowledge...."

L64-6: What about genetic studies? These have revealed information about migratory behavior, etc. Please include.

L74: "an excellent tool to answer these questions". Reword to something a little more focused on your coming questions e.g., "provide/present a viable/robust tool for understanding the seasonal and diurnal calling behavior of SRW". This then links into your next paragraph.

L76-80: Q1 is basically revealed by Q2 so I don't see why this is a separate focus. PAM is important and will reveal presence. Again, is this paper focused on PAM or on the acoustic ecology of the species? My feeling is the latter and PAM is the tool to get you to the important questions about your species.

Methods

L86: delete "of".

L103: i.e., a 12.5% duty cycle. Please add. Please also mention that a 4 kHz sampling rate covers the vocalisation range of this species.

L105-12: Please include the specific depth at the deployment location and the depth of the recorder in the water column.

L115-9: Your classification methods need to be expanded. So, you first manually scanned all recordings for call presence. Does this include all calls no matter the SNR? If so, please state this or the SNR you used as a cut off. Once a call was identified, you classified the call type based on a previous quantitative classification of a subset of your data. Please include this information. From here you then counted the number of calls and call types per recording/hour/day/month? More information is required.

L122: all acoustic recordings or a subset? Please clarify.

L136: what about missed detections? How many calls were missed by the detector? There is no Table in Results detailing number of calls manually classified, classified by detector, missed by detector, missed by human, etc. Please add such a Table. [I see information is provided in the SI but this should be summarised for the reader in a Table].

L163: delete "Program".

L170: "calls" should be "call".

L172: More information is required about the difference between positive and negative values. "This correction resulted in a mean adjusted call rate where positive values indicate....".

Results

Footnote 1 pg 9: remove "a" from the first sentence.

L200: This is half of a sentence. I suggest including a statement after this along the lines of "The vocalisation rate increased sharply in May (41, 2.9), coinciding with ...[winter migration into the area]."

I would like to see a Figure with spectrograms of the five major call types for the generalist reader. Corresponding audio files would also be nice. Especially for the upcall, which you run and auto detector on. Ideally, at least 5 examples of different upcalls would be good to see what variability in call is allowable to be classified by the detector (and human as an upcall).

L245-57: This information should be presented in a clear Table outlining the detector's success (or failure). Number of calls manually detected, Number of calls autodetected, number of missed calls, number of false detections - all with actual counts and percentages. Given this is then discussed by month I suggest then breaking it down by month too. This was run on 10% of the

data – was this spread across all months? Please include more information in the methods at L133 to confirm this.

Discussion

After the focus of the paper is clarified (PAM or acoustic ecology), the Discussion can be tightened to reflect the focus. I suggest adding a concluding paragraph to leave the reader with a clear take home message.

Figures and Tables

There are too many figures.

Figure 2: delete figure as the information is explained in the text and covered in Figures 3 & 4.

Delete Figure 4 as Figure 5 provides more information. If the authors feel information from Fig. 4 needs to be included, this should be presented in a Table of call numbers.

Figure 6 & 7: more information about what positive and negative numbers mean is required. Each Figure should be stand-alone and not require consultation with the main text.

Author's Response to Decision Letter for (RSOS-181487.R0)

See Appendix B.

Decision letter (RSOS-181487.R1)

12-Feb-2019

Dear Dr Webster,

I am pleased to inform you that your manuscript entitled "Temporal variation in the vocal behaviour of southern right whales in the Auckland Islands, New Zealand" is now accepted for publication in Royal Society Open Science.

on behalf of Dr Asha de Vos (Associate Editor) and Kevin Padian (Subject Editor)
openscience@royalsociety.org

Associate Editor Comments to Author (Dr Asha de Vos):

Thank you for working hard to incorporate the valuable comments. The paper is now a better version of an already great and important paper. Thank you!

Appendix A**ROYAL SOCIETY
OPEN SCIENCE****Temporal variation in the vocal behaviour of southern right
whales in the Auckland Islands, New Zealand**

Journal:	Royal Society Open Science
Manuscript ID	RSOS-181487
Article Type:	Research
Date Submitted by the Author:	05-Sep-2018
Complete List of Authors:	Webster, Trudi; University of Otago, Department of Marine Science; Otago Museum, Van Parijs, Sofie; National Oceanic and Atmospheric Administration Rayment, Will; University of Otago, Department of Marine Science Dawson, Steve; University of Otago , Department of Marine Science
Subject:	behaviour < BIOLOGY, ecology < BIOLOGY, Acoustics < PHYSICS
Keywords:	acoustic behaviour, passive acoustic monitoring, vocalisations, diel variation, seasonality, right whales
Subject Category:	Biology (whole organism)

**Temporal variation in the vocal behaviour of southern right whales in the Auckland**
**Islands, New Zealand**

Webster TA^{1,2}, Van Parijs SM³, Rayment WJ¹ and Dawson SM¹

¹ *Department of Marine Science, University of Otago, 310 Castle Street, Dunedin 9016,*
*New Zealand*

² *Otago Museum, 419 Great King Street, Dunedin 9016, New Zealand.*

³ *Protected Species Branch, Northeast Fisheries Science Center, 166 Water Street,*
*Woods Hole, Massachusetts 02543*

Corresponding author: Trudi Webster, science@yeptrust.org.nz

Short title: Vocal behaviour of right whales

**Abstract**

Autonomous recorders are frequently utilised for examining vocal behaviour of animals, and
are particularly effective in remote habitats. Here a system was employed to examine whether
the acoustic behaviour of southern right whales differed seasonally and throughout the day at
their main calving ground in New Zealand. An autonomous recorder was moored at the
isolated sub-Antarctic Auckland Islands for a year. Recordings were made in each month
except June, and vocalisations were audible in all months with recordings except January. A
total of 35,487 calls were detected, of which *upcalls* were the most common (11,623). Call
rate peaked in August (288 ± 5.9 [SE] calls/hour) and July (194 ± 8.3). Vocal behaviour varied
diurnally with highest call rates at dusk and night, consistent with the concept that *upcalls*
function primarily as contact calls. Zero-inflated model results confirmed that seasonal
variation was the most important factor for explaining differences in vocal behaviour. An
automated detector designed to expedite the analysis process for North Atlantic right whales
correctly identified 80% of *upcalls*, although false detections were frequent, particularly
when call rates were low. This study is the first to attempt year-round monitoring of southern
right whale presence in New Zealand.

**Keywords**

Acoustic behaviour, passive acoustic monitoring, vocalisations, diel variation, seasonality,
right whales

**Background**

Animals vary their behaviour in response to a wide range of factors, including changes in
light levels [1], availability of prey [2], predation pressure [3,4] and weather conditions [5,6].
Similarly, vocal behaviour of many animals changes temporally. For example, terrestrial
species, such as wolves (*Canis lupus* [7]) and many temperate and tropical bird species [8],
exhibit higher levels of vocal activity during the late evening and dawn. The drivers of
temporal variation in vocal behaviour can be difficult to determine, but may relate to optimal
ambient noise and propagation conditions [9], energy reserves [10], predation risk [11] or
sociality [8,12]. The calling activity of frogs [13] and some insects [14,15] is correlated with
precipitation and temperature.

Marine mammals in general, and cetaceans in particular, are highly vocal, relying on sound
for communication [16], navigation [17] and feeding [18]. Consequently, the occurrence of
species-specific vocalisations can reveal information about a species' presence. For example,
in the Gulf of Alaska, echolocation clicks made by sperm whales (*Physeter macrocephalus*)
are detected more often in summer than in winter, consistent with known seasonal migration
of reproductive males from productive high latitudes in summer, to the lower latitudes
favoured by breeding females in winter [19]. Vocalisation types and rates can change over
short timescales (hours/days) or longer monthly or seasonal timescales revealing details of a
species' behaviour. Blue whales (*Balaenoptera musculus*) from the eastern tropical Pacific
show higher vocalisation rates between February and June; thought to be associated with
their foraging behavior which is also highly seasonal [20]. Diel changes in acoustic behaviour
have been observed in both mysticetes [21,22] and odontocetes [23,24]. For example, sound
pressure levels of vocalisations of wintering humpback whales (*Megaptera novaeangliae*) off

Maui increase at sunset, remain relatively high through the night and progressively decrease
from sunrise [25]. Vocal behaviour can also be used to infer the level and type of activity that
an animal is engaged in. Elevated echolocation rates and buzzes (associated with feeding)
indicate that foraging at night is the norm for several odontocete species, including harbour
porpoise (*Phocoena phocoena* [26]), finless porpoise (*Neophocoena asiaorientalis* [24]) and
Risso's dolphin (*Grampus griseus* [23]).

Passive acoustic monitoring (PAM) has become an increasingly important research tool,
particularly for documenting a species' presence [19,21] and how vocal behaviour changes
with time [20]. Autonomous recording systems used to detect species-specific calls and
monitor temporal variation in vocal behaviour have been fundamental to the establishment of
protection measures for North Atlantic right whales (NARWs, *Eubalaena glacialis*) in and
around shipping lanes [27]. Typically, passive acoustic detection and localisation of right
whales uses *upcalls* [28,29] which are the most common call type [30,31,32], and are
produced by both males and females [33,34]. PAM is increasingly used to monitor right
whales, particularly in remote regions or those with inhospitable weather conditions
[35,36,37]. Consequently, automated detection systems [28] have been developed to facilitate
processing the large quantities of data produced by PAM.

Most knowledge of the behaviour of southern right whales (SRWs, *Eubalaena australis*) in 45
New Zealand comes from visual surveys and observations at the sub-Antarctic Auckland
Islands [38,39,40,41]. Right whales are the only cetacean species known to be regularly
present in Auckland Islands waters [42] and arrive there  each winter to calve [38]. Practical
constraints restrict the duration of research expeditions to these remote islands; beyond a

wintertime window of up to eight weeks [39] little is known about how or when right whales
use this habitat. The vocal repertoire of SRWs at the Auckland Islands has received attention
recently [43], but we do not know how long SRWs are present in this habitat or how their
vocal behaviour varies over time. Given the limitations of the remote location and inclement
weather conditions in the sub-Antarctic, autonomous acoustic recorders [44] provide an
excellent tool to answer these questions.

This research addressed three specific questions: (1) what do autonomous acoustic recorders

[revised manuscript text omitted]

Perrin, W.F., Würsig, B. and Thewissen, J.G.M. (eds.) 2009. *Encyclopedia of marine*
*mammals*. Academic Press.
59. Rayment W, Webster T, Brough T, Jowett T, Dawson S 2018 Seen or heard? A
comparison of visual and acoustic autonomous monitoring methods for investigating
temporal variation in occurrence of southern right whales. *Mar. Biol.* **165**,
12 (<https://doi.org/10.1007/s00227-017-3264-0>)

60. Urazghildiiev IR, Parks SE 2014 Objective classification of North Atlantic right whale
(*Eubalaena glacialis*) vocalizations to improve passive acoustic detection (No. e322v1).
*PeerJ PrePrints* (<http://dx.doi.org/10.7287/peerj.preprints.322v1>)
61. Clark CW, Clark JM 1980 Sound playback experiments with southern right whales.
*Science* **207**, 663-665 (<https://doi.org/10.1126/science.207.4431.663>)
62. Parks SE, Hamilton PK, Kraus SD, Tyack PL 2005 The gunshot sound produced by male
North Atlantic right whales (*Eubaleana glacialis*) and its potential function in
reproductive advertisement. *Mar. Mamm. Sci.* **21**, 458-475.
(<https://doi.org/10.1111/j.1748-7692.2005.tb01244.x>)
63. Matthews LP, McCordic JA, Parks SE 2014 Remote acoustic monitoring of North Atlantic
right whales (*Eubalaena glacialis*) reveals seasonal and diel variations in acoustic
behavior. *PLoS ONE* **9**, e91367. (<https://doi.org/10.1371/journal.pone.0091367>)
64. Best PB 1994 Seasonality of reproduction and the length of gestation in southern right
whales *Eubalaena australis*. *J. Zool.* **232**, 175-189. (<https://doi.org/10.1111/j.1469->
[7998.1994.tb01567.x](https://doi.org/10.1111/j.1469-7998.1994.tb01567.x))
65. Torres LG et al. 2016 Demography and ecology of southern right whales *Eubalaena*
*australis* wintering at sub-Antarctic Campbell Island, New Zealand. *Polar Bio.* 1-12.
(<https://doi.org/10.1007/s00300-016-1926-x>)
66. Office of Naval Research. 1997 Northern right whale monitoring project: Final report.
Available from Office of Naval Research, Arlington. 66 pp.
67. Baumgartner MF, Lysiak NS, Schuman C, Urban-Rich J, Wenzel FW 2011 Diel vertical
migration behavior of *Calanus finmarchicus* and its influence on right and sei whale
occurrence. *Mar. Ecol. Prog. Ser.* **423**, 167-184. (<https://doi.org/10.3354/meps08931>)

68. Corkeron PJ, Connor RC 1999 Why do baleen whales migrate? *Mar. Mamm. Sci.* **15**,
1228-1245 (<https://doi.org/10.1111/j.1748-7692.1999.tb00887.x>)
69. Parks SE, Hotchkin CF, Cortopassi KA, Clark CW 2012 Characteristics of gunshot sound
displays by North Atlantic right whales in the Bay of Fundy. *J. Acoust. Soc. Am.* **131**,
3173-3179. (<https://doi.org/10.1121/1.3688507>)
70. Schmidt KA, Belinsky KL 2013 Voices in the dark: predation risk by owls influences dusk
singing in a diurnal passerine. *Behav. Ecol. Sociobiol.* **67**, 1837-1843.
(<https://doi.org/10.1007/s00265-013-1593-7>)
71. Baumgartner MF, Mussoline SE 2011 A generalized baleen whale call detection and
classification system. *J. Acoust. Soc. Am.* **129**, 2889-2902
(<https://doi.org/10.1121/1.3562166>)

Location of the DSG-Ocean recorder for southern right whale vocalisations in Laurie Harbour, Port Ross, sub-Antarctic Auckland Islands

	2011						2012						
	Jul	Aug	Sep	Oct	Nov	Dec	Jan	Feb	Mar	Apr	May	Jun	Jul
DSG-Ocean deployment	[Dark grey shading]												
Successful recordings	[Mid grey shading]												
SRW calls present	[Light grey shading]							[Light grey shading]					

Timetable of recorder deployment in Laurie Harbour, Port Ross. Dark grey represents deployment duration, mid grey is recording duration and light grey is presence of whale calls

Mean number of all calls (black) and upcalls (grey) per day for southern right whales at the Auckland Islands. N.B.: no data were collected from 2-5 August 2011 (inclusive).

	2011						2012					
	July	Aug	Sept	Oct	Nov	Dec	Jan	Feb	Mar	Apr	May	
All calls	Shaded							Shaded				
Upcall	Shaded							Shaded				
Downcall	Shaded							Shaded				
Gunshots	Shaded											
Tonal low	Shaded											
High	Shaded											
Hybrid	Shaded											
Very high	Shaded											
Pulsive	Shaded											
Blow	Shaded											

Presence/absence of different calls types for right whales recorded at the Auckland Islands between July 2011 and May 2012. Shaded bars indicate presence

Mean call rate (calls h⁻¹ ±SE) for the five most common call types (upcall, downcall, gunshot, tonal low and pulsive) in each month for southern right whales recorded at the Auckland Islands between July 2011 and May 2012.

Mean adjusted hourly vocalisation rate (calls h⁻¹ ±SE) for southern right whales recorded at the Auckland Islands between July 2011 and May 2012. Rates are adjusted to compensate for variation in call rate throughout the year.

Mean adjusted call rate (calls $\text{h}^{-1} \pm \text{SE}$) per diel period for southern right whales recorded at the Auckland Islands between July 2011 and May 2012. Rates are adjusted to compensate for variation in call rate throughout the year.

Mean call rate (calls h⁻¹ ±SE) for the five most common call types (upcall, tonal low, pulsive downcall and gunshot) in each diel period for southern right whales recorded at the Auckland Islands between July 2011 and May 2012.

The number of automatically detected upcalls per recording vs the number of upcalls detected by the analyst. Right whale vocalisations were recorded at the Auckland Islands between July 2011 and May 2012. Best fit linear regression is shown ($y = 0.6743x - 0.1261$, $r^2 = 0.76$). Points have been jittered by 0.6 to offset and reveal overlapping points.

	ALL CALLS				UPCALLS ONLY			
Count model coefficients								
Variable	Estimate	SE	z value	Pr(> z)	Estimate	SE	z value	Pr(> z)
Season Spring	0.05	0.05	0.92	0.36	0.08	0.06	1.43	0.15
Season Summer	-0.54	0.28	-1.91	0.06	-0.19	0.27	-0.69	0.49
Season Winter	0.64	0.05	12.69	<2.00 x 10 ⁻¹⁶	0.26	0.06	4.76	1.92 x 10 ⁻⁵
Diel Period Dawn	0.02	0.08	0.32	0.75	-0.14	0.09	-1.61	0.11
Diel Period Dusk	0.28	0.07	3.99	6.52 x 10 ⁻⁵	0.3	0.08	3.79	1.48 x 10 ⁻⁴
Diel Period Morning	-0.13	0.05	-2.62	0.01	-0.01	0.06	-0.26	0.79
Diel Period Night	0.25	0.04	6.27	3.66 x 10 ⁻¹⁰	0.24	0.04	5.34	9.33 x 10 ⁻⁸
Zero-inflation model coefficients								
Season Spring	-1.61	0.06	-24.85	<2.00 x 10 ⁻¹⁶	-1.32	0.07	-18.19	<2.00 x 10 ⁻¹⁶
Season Summer	3.65	0.32	11.35	<2.00 x 10 ⁻¹⁶	3.35	0.32	10.39	<2.00 x 10 ⁻¹⁶
Season Winter	-4.8	0.11	-44.68	<2.00 x 10 ⁻¹⁶	-3.49	0.08	-42.2	<2.00 x 10 ⁻¹⁶
Diel Period Dawn	0	0.13	-0.02	0.99	-0.01	0.14	-0.06	0.95
Diel Period Dusk	-0.47	0.12	-3.79	1.49 x 10 ⁻⁴	-0.49	0.13	-3.93	8.62 x 10 ⁻⁵
Diel Period Morning	0.13	0.08	1.63	0.1	0.15	0.08	1.83	0.07
Diel Period Night	-0.66	0.07	-9.5	<2.00 x 10 ⁻¹⁶	-0.68	0.07	-9.56	<2.00 x 10 ⁻¹⁶

Results of the zero-inflated model examining seasonal and diel differences in all call and upcall presence and rate for right whales recorded at the Auckland Islands. The reference level for diel period is afternoon and the reference level for season is Autumn. Significant results are highlighted in grey.

Appendix B

MS Reference Number: RSOS-181487

Article Status: SUBMITTED

MS Dryad ID: RSOS-181487

MS Title: Temporal variation in the vocal behaviour of southern right whales in the Auckland Islands, New Zealand

MS Authors: Webster, Trudi; Van Parijs, Sofie; Rayment, Will; Dawson, Steve

Contact Author: Trudi Webster

Contact Author Email: science-advisor@yeptrust.org.nz

Response to Referees

Associate Editor's comments (Dr Asha de Vos):

Associate Editor: 1

Comments to the Author:

This manuscript will be a valuable contribution to the literature after the suggested revisions. Thank you for your submission and we hope you will be able to address the reviewers concerns in a timely manner.

Reviewers' Comments to Author:

Reviewer: 1

Comments to the Author(s)

This is a really nice study. My only main comment is that it would be helpful for the reader to understand how this habitat relates to the bigger picture of SRW distribution and knowledge. I've included some minor comments within the attached document.

Response: further information about how this habitat relates to the bigger picture of SRW distribution and knowledge has been included and is addressed in response to the comment on line 64 (see below).

Line 64: *I think it would be useful to give the reader a broader understanding of SRW population and distribution for this stock and to describe how / why this study is important in the bigger context. How can it inform what is and isn't known about this stock?*

Response to line 64: The following has been added to give a broader introduction to SRW and help to put this study into context. "Southern right whales (SRWs, *Eubalaena australis*) have a circumpolar distribution in the southern hemisphere, with major breeding populations in South Africa, Argentina, Brazil, Australia and New Zealand. SRWs were hunted to the brink

of extinction in New Zealand [38], but since the cessation of whaling the population has grown at an estimated 7% resulting in a recent abundance estimate of 2169 (95%: 1836-2563) [39]. For other SRW populations mainland calving grounds mean that they are easily observable. SRWs in New Zealand only calve in the remote sub-Antarctic Auckland Islands, so we know very little about distribution and behaviour through the year.” See lines 64-71.

Line 67: arrive from where or describe it is or isn't known. Also is this the only purpose of this habitat or are there non m/c whales there too? It might be an important descriptor to put these calls in context.

Response to line 67: “from offshore feeding grounds” has been added to address from where the whales arrive. An additional comment has been added to explain that non m/c whales are also present “Non-breeding whales are also present potentially due to social factors [41]”. See line 75.

Line 86: Remove “of”

Response to line 86: “of” has been removed (line 94).

Line 141: I'm struggling with the second value and how it is calculated. Needs a little more explanation of the process for that – how do you know what was correctly assigned in the TPR process or does it come from the PPR? It's confusing as written.

Response to line 141: Agree that this is confusing as currently written. The sentence paragraph has been re-written as “The second measure (True Positive Rate, TPR) is the percentage of *upcalls* detected by the analyst also logged by the detector; the converse enabled estimation of false negatives.” See lines 150-151.

Line 156: to nautical dawn?

Response to line 156: “dawn” has been corrected to “nautical dawn”. See line 165.

Line 170: I'm having a hard time understanding how this would actually work. Are you only doing this for a subset of days with atypically high calling rates?

Response to line 170: This is done for all days, which has now been clarified in the text by adding “for each day”. For further explanation of how mean adjusted call rate is calculated “(e.g. hour or diel period)” was also added. The response to a comment (line 172) from reviewer 2 also further explains what this calculation means. See lines 178-180.

Line 255: What proportion? This seems important to understand.

Response to line 255: This was a large proportion of calls, exact numbers are not available as these data were not explicitly collected as part of this study. We have added “large” for further clarification. See line 266-267.

Line 295: Also could be the result of comparing different sorts of habitats

Response to line 295: “or because whales were sampled in different habitats” has been added. See lines 306-307.

Line 324: Make it clear that this is all RW species

Response to line 324: “all right whale species” has been added for clarification. See lines 313-315.

Line 374: It might be important to mention what can be learned from acoustic recorders and their utility really depends on the question one is trying to answer. Down the road of there is clearer linkage of acoustics with whale behaviour which you have sort of touched on, ie if the acoustic outputs in a calving area are somehow a lot different than feeding and/or mating area and that could be better described then maybe acoustic

Response to line 374: A sentence has been added to the discussion to highlight the linkage with acoustics and whale behaviour “Furthermore, different call types might be indicative of different behaviours which may not be apparent from visual observation alone”. See lines 385-386.

Reviewer: 2

Comments to the Author(s)

This manuscript examines the seasonal and diel acoustic behaviour of southern right whales at a remote calving ground using PAM. The paper jumps between focusing on PAM and SRW acoustic ecology, which leads to a lack of conceptual flow. A little more cohesion between the concepts will help the manuscript flow better (e.g., acoustic ecology of a difficult to study species is revealed by PAM). A thorough read of the manuscript is needed to ensure the grammatical errors throughout are remedied. Some (but by no means all) are noted in the specific comments below. There are too many Figures presented with the majority replicating information and can be removed from the manuscript. What is missing, however, is a clear Table detailing the auto-detector success/failure rates. After major revision the manuscript will be an important contribution to the literature.

Response: Further clarification throughout the manuscript, of the focus on acoustic ecology of SRW as revealed by PAM, has been added as addressed in the specific comments below. Additional grammatical errors have been remedied, figures reduced and a table added as suggested (see specific

comments below).

Specific comments are provided below:

Abstract

L3-6: I suggest combining these two sentences, and adding in a sentence about southern right whales (e.g., distribution, ecology, etc.). Your opening paragraph in the Introduction focuses on behaviour and yet your abstract focuses on PAM.

Response to L3-6: The two sentences have been joined as follows “A recorder was moored at the isolated sub-Antarctic Auckland Islands for a year to examine whether the acoustic behaviour of southern right whales differed seasonally and throughout the day at their main calving ground in New Zealand” There is a 200 word limit for the abstract so a short sentence about SRWs was added “Southern right whales are known to have an extensive acoustic repertoire”. See lines 3-6.

L10: add “detected” after call rates.

Response to L10: “detected” has been added. See line 10.

Background

L22-30: I suggest including the word “[diurnal” in this paragraph – as that is what you are describing.

Response to L22-30: Diurnal has been added in this paragraph. Which now reads “...exhibit diurnal variation in vocal activity with higher call rates during the late evening and dawn”. See lines 24-27.

*L54: Add “For example,” autonomous recordings....
This will allow your sentences to link better.*

Response to L54: For example has been added. See line 54.

L64: You need a link to the previous paragraph, for example: “In the Southern Hemisphere, most knowledge....”

Response to L64: Following a comment from reviewer 1 (line 64) a paragraph on SRW has been added to give the reader the bigger picture. This serves as a linkage between the two paragraphs, and thus makes this addition unnecessary. See lines 64-71.

L64-6: What about genetic studies? These have revealed information about migratory behavior, etc. Please include.

Response to L64-6: A genetic reference (Carroll et al. 2013 [39]) has been added as part of the broader picture of SRW (see previous comment on line 64 from reviewer 1). See line 68.

L74: *“an excellent tool to answer these questions”. Reword to something a little more focused on your coming questions e.g., “provide/present a viable/robust tool for understanding the seasonal and diurnal calling behavior of SRW”. This then links into your next paragraph.*

Response to L74: The text has been changed to “present a robust tool for understanding the seasonal and diurnal calling behaviour of SRW”. See lines 80-82.

L76-80: *Q1 is basically revealed by Q2 so I don't see why this is a separate focus. PAM is important and will reveal presence. Again, is this paper focused on PAM or on the acoustic ecology of the species? My feeling is the latter and PAM is the tool to get you to the important questions about your species.*

Response to L76-80: We agree that the focus of the paper is the acoustic ecology of SRW and PAM is a tool to help with this. Q1 is about seasonal presence, Q2 is about vocal behaviour. Q1 has been reworded to reflect this focus “what is the seasonal variation in the presence of whales at the Auckland Islands as revealed by autonomous acoustic recorders”. Rates and types has been added to Q2 to provide further clarification “how do call rates and types vary temporally”. See lines 84-88.

Methods

L86: *delete “of”.*

Response to L86: “of” has been deleted (line 94).

L103: *i.e., a 12.5% duty cycle. Please add. Please also mention that a 4 kHz sampling rate covers the vocalisation range of this species.*

Response to L103: “(i.e. a 12.5% duty cycle)” and “(covering the vocalisation range of this species)” have been added to the text. See lines 109-111.

L105-12: *Please include the specific depth at the deployment location and the depth of the recorder in the water column.*

Response to L105-12: The deployment depth has been made more specific “16-20 m” instead of <20m. The depth of the recorder is covered in line 109, but we have added “on the seabed” to make it clear that the depth is 1m from the seabed. See lines 96 and 115.

L115-9: *Your classification methods need to be expanded. So, you first manually scanned all recordings for call presence. Does this include all calls*

no matter the SNR? If so, please state this or the SNR you used as a cut off. Once a call was identified, you classified the call type based on a previous quantitative classification of a subset of your data. Please include this information. From here you then counted the number of calls and call types per recording/hour/day/month? More information is required.

Response to L115-9: “regardless of signal to noise ratio” has been added to make this clearer. We have also added “based on a previously determined quantitative classification system described in a concurrent study of this population” to further clarify what the classification was based on. Details about the number of calls and call types per time period is detailed later in the methods (under the section on statistical analyses). Lines 124-128.

L122: *all acoustic recordings or a subset? Please clarify.*

Response to L22: “All” has been added for clarification. See line 131.

L136: *what about missed detections? How many calls were missed by the detector? There is no Table in Results detailing number of calls manually classified, classified by detector, missed by detector, missed by human, etc. Please add such a Table. [I see information is provided in the SI but this should be summarised for the reader in a Table].*

Response to L136: This information was summarised in the paragraph on effectiveness of the automated detector (in the results section). We have added an additional table (Table 2) to summarise the detection effectiveness information as suggested. The following data have been included: number of calls manually detected, number of upcalls autodetected, number of true positive calls, number of false positive calls, number of false negative calls, PPV and TPR. These have been broken down for each month separately, except for October to April where data were combined due to a low call rate. See lines 655-660.

L163: *delete “Program”.*

Response to 163: “Program” has been deleted (line 172).

L170: *“calls” should be “call”.*

Response to L170: “Calls” has been changed to “call” (line 178).

L172: *More information is required about the difference between positive and negative values. “This correction resulted in a mean adjusted call rate where*

positive values indicate....”.

Response to L172: We have added further information to the text to explain this better. “where positive values indicate an above average (and negative a below average) rate during a particular time period allowing for seasonal variation in the total number of calls detected.”

Results

Footnote 1 pg 9: remove “a” from the first sentence.

Response to Footnote1: “a” has been deleted.

L200: This is half of a sentence. I suggest including a statement after this along the lines of “The vocalisation rate increased sharply in May (41, 2.9), coinciding with[winter migration into the area].”

I would like to see a Figure with spectrograms of the five major call types for the generalist reader. Corresponding audio files would also be nice. Especially for the upcall, which you run and auto detector on. Ideally, at least 5 examples of different upcalls would be good to see what variability in call is allowable to be classified by the detector (and human as an upcall).

Response to L200: The coincidence with the winter migration is discussed in the discussion. We would prefer to leave this for the discussion as it is interpretation of the results. Figure 2 showing all of the major call types has been added to the manuscript as suggested to provide additional classification information for the reader.

L245-57: This information should be presented in a clear Table outlining the detector’s success (or failure). Number of calls manually detected, Number of calls autodetected, number of missed calls, number of false detections – all with actual counts and percentages. Given this is then discussed by month I suggest then breaking it down by month too. This was run on 10% of the data – was this spread across all months? Please include more information in the methods at L133 to confirm this.

Response to L245-57: A table (Table 2) has been included to summarise the detection information as suggested in a previous comment (line 136). The detector was run on all of the data but only 10% of the data were analysed in detail, the 10% was randomly selected but this does include representation for each of the months (although samples are not evenly spread). We have added “which included representation of each month” to give further detail in the text (line 142).

Discussion

After the focus of the paper is clarified (PAM or acoustic ecology), the Discussion can be tightened to reflect the focus. I suggest adding a concluding paragraph to leave the reader with a clear take home message.

Response to discussion: The focus of the paper has been clarified as acoustic ecology, and a concluding paragraph has been added to reflect this and leave the reader with a clear take home message. "In this study we used PAM to investigate the acoustic ecology and behaviour of SRW at their only known calving ground in New Zealand. SRWs in particular lend themselves to PAM due to their highly vocal nature and extensive vocal repertoire. The year-round monitoring of SRWs at the Auckland Islands expanded upon our current knowledge, revealing that they are at least occasionally present in every month of the year. Call rates in winter were however four times higher than any other season, suggesting seasonal variations in density as a result of winter migration inshore to calve. *Upcalls* were the most commonly produced call type proving ideal for automated detection throughout the year. Seasonality was apparent in many other call types, likely indicating a change of behaviour during winter. Call rates exhibited a clear diurnal pattern with higher vocal activity during dusk and night time potentially because reduced visual contact increases the need for acoustic communication. This study begins to scratch the surface of temporal variation in the acoustic behaviour of SRWs, but there is much more to be revealed." See lines 389-400.

Figures and Tables

There are too many figures.

Response to figures and tables: Two figures have been deleted as suggested (see specific comments below).

Figure 2: delete figure as the information is explained in the text and covered in Figures 3 & 4.

Response to Figure 2: Figure 2 has been deleted and numbering of figures altered accordingly.

Delete Figure 4 as Figure 5 provides more information. If the authors feel information from Fig. 4 needs to be included, this should be presented in a Table of call numbers.

Response to Figure 4: Figure 4 has been deleted and numbering of figures altered accordingly. Additional text has been added so that the detail from figure 4 is not lost. "*Hybrid, high and gunshot* vocalisations exhibited similar patterns of seasonal presence and are completely absent between December

and April” and “were recorded much less frequently than *upcalls*, and in some months not at all (particularly during summer)”

Figure 6 & 7: *more information about what positive and negative numbers mean is required. Each Figure should be stand-alone and not require consultation with the main text.*

Response to Figure 6 & 7: Additional information has been added to the captions for figures now 5 & 6 to clarify “Positive values indicate an above average rate and negative values indicate a below average rate”. See lines 678-679, 683-684.